# Quorum-Sensing Signal DSF Inhibits the Proliferation of Intestinal Pathogenic Bacteria and Alleviates Inflammatory Response to Suppress DSS-Induced Colitis in Zebrafish

**DOI:** 10.3390/nu16111562

**Published:** 2024-05-22

**Authors:** Ruiya Yi, Bo Yang, Hongjie Zhu, Yu Sun, Hailan Wu, Zhihao Wang, Yongbo Lu, Ya-Wen He, Jing Tian

**Affiliations:** 1Key Laboratory of Resource Biology and Biotechnology in Western China, Ministry of Education, College of Life Sciences, Northwest University, Xi’an 710069, China; yiruiya@stumail.nwu.edu.cn (R.Y.); 202221238@stumail.nwu.edu.cn (B.Y.); 202032724@stumail.nwu.edu.cn (H.Z.); 202121675@stumail.nwu.edu.cn (Y.S.); wuhailann@stumail.nwu.edc.cn (H.W.); 20229029@nwu.edu.cn (Z.W.); luyongbo@stumail.nwu.edu.cn (Y.L.); 2State Key Laboratory of Microbial Metabolism, Joint International Research Laboratory of Metabolic & Development Sciences, School of Life Sciences & Biotechnology, Shanghai Jiao Tong University, Shanghai 200240, China

**Keywords:** diffusible signal factor (DSF), quorum-sensing signal, inflammatory bowel disease (IBD), zebrafish model, gut microbiota

## Abstract

The imbalance of gut microbiota is an important factor leading to inflammatory bowel disease (IBD). Diffusible signal factor (DSF) is a novel quorum-sensing signal that regulates bacterial growth, metabolism, pathogenicity, and host immune response. This study aimed to explore the therapeutic effect and underlying mechanisms of DSF in a zebrafish colitis model induced by sodium dextran sulfate (DSS). The results showed that intake of DSF can significantly improve intestinal symptoms in the zebrafish colitis model, including ameliorating the shortening of the intestine, reducing the increase in the goblet cell number, and restoring intestinal pathological damage. DSF inhibited the upregulation of inflammation-related genes and promoted the expression of claudin1 and occludin1 to protect the tightness of intestinal tissue. The gut microbiome analysis demonstrated that DSF treatment helped the gut microbiota of the zebrafish colitis model recover to normal at the phylum and genus levels, especially in terms of pathogenic bacteria; DSF treatment downregulated the relative abundance of *Aeromonas hydrophila* and *Staphylococcus aureus*, and it was confirmed in microbiological experiments that DSF could effectively inhibit the colonization and infection of these two pathogens in the intestine. This study suggests that DSF can alleviate colitis by inhibiting the proliferation of intestinal pathogens and inflammatory responses in the intestine. Therefore, DSF has the potential to become a dietary supplement that assists in the antibiotic and nutritional treatment of IBD.

## 1. Introduction

As a common chronic recurrent inflammatory gastrointestinal disease, inflammatory bowel disease (IBD) is mainly manifested as abdominal pain, diarrhea, bleeding, and weight loss [1,2]. Clinically, IBD comprises two main subtypes: ulcerative colitis (UC), which involves persistent inflammation of the colonic mucosa, and Crohn’s disease (CD), which exhibits discontinuous damage along the gastrointestinal tract [3]. In the past few decades, the global incidence rate of IBD has risen rapidly, which has become a serious public health problem and brought a heavy economic burden to families and society [4]. As a lifelong disease, the etiology of IBD is still largely unknown, but its pathogenesis is generally considered to involve complex interactions among genetic susceptibility, gut microbiota, immune responses, and environmental and lifestyle factors [4,5,6]. So far, there is no method that can completely cure IBD [7]. Therefore, finding new and more efficient treatments for IBD remains an urgent task.

The gut microbiota coexists and co-evolves with the host, exerting beneficial effects on the host, such as forming an intestinal barrier to prevent pathogen invasion, assisting the host in necessary metabolism and nutrition, and promoting immune system maturation [8]. Emerging evidence implies that the gut microbiota is an important factor leading to the development of IBD. For example, early microbial exposure can affect the development and maturation of the intestinal immune system in humans and mice and have an impact on susceptibility to IBD later on [9]. The gut microbiota of IBD patients exhibits various abnormal phenotypes, such as decreased diversity, reduced anti-inflammatory bacteria, and increased pro-inflammatory bacteria. The overgrowth of pathogenic bacteria such as invasive *Escherichia coli*, *Staphylococcus*, and *Pseudomonas* can cause endotoxin translocation, leading to various infections, destruction of the intestinal epithelium, increased expression of pro-inflammatory factors, and exacerbation of intestinal inflammation [10]. The excessive growth of pathogenic bacteria in the intestine can also compete for intestinal nutrients and the habitation space of probiotics, leading to malnutrition of the intestinal mucosa and destruction of microbial balance. In addition, it will also degrade intestinal mucins and disrupt tight junctions, thereby increasing intestinal permeability [11]. Therefore, inhibiting intestinal infection or the proliferation of intestinal pathogens has become an important strategy for the treatment of IBD.

The quorum sensing (QS) system is a communication method among bacteria that functions through the secretion of small molecule signals to coordinate group behaviors and adapt to different environmental conditions [12]. QS plays a crucial role in regulating the production of bacterial virulence factors and biofilm formation, which makes it closely related to diseases caused by bacteria; utilizing natural QS molecules to regulate intestinal microbiota imbalance is a new strategy for treating IBD [13]. Related studies have shown that the QS signal *N*-acyl homoserine lactone (AHL) in the human body participates in the interaction between the host and microbiota in IBD, exerting potential beneficial effects in inhibiting inflammatory responses and maintaining intestinal barrier integrity [14,15]. Additionally, another QS signal, autoinducer-2 (AI-2), plays an essential role in the adhesion and colonization of probiotics on the intestinal epithelium [12]. Exogenous AI-2 can alleviate intestinal inflammation and reverse dysbiosis in a necrotizing enterocolitis mouse model, mitigating antibiotic-induced intestinal microbiota imbalance [16,17]. The DSF family signal is a QS signal commonly present in Gram-negative bacteria [18], and its family member, *cis*-2-hexadecenoic acid (c2-HDA), inhibits the expression of *Salmonella* virulence genes in mouse colitis models [19]; *cis*-2-dodecaenoic acid (BDSF) inhibits the expression of *Candida albicans* adhesion factors and inhibits vaginitis infection [20]. In addition, our previous studies have shown that *cis*-11-methyldecenoic acid (DSF) has a restorative effect on the inflammatory response induced by bacterial endotoxins [21]. At present, the impact of DSF on gut microbiota and inflammatory status in IBD has not been reported.

In the present study, the intervention effect of DSF on zebrafish colitis was evaluated by establishing a zebrafish IBD model induced by dextran sodium sulfate (DSS). Meanwhile, using 16s rRNA sequencing, we investigated the regulation of DSF on the structure and composition of colonic microbiota in colitis from the perspective of the gut microbiota. This study will reveal the possibility of utilizing bacterial QS signals to regulate the imbalance of gut microbiota and optimize the ecological structure of intestinal microorganisms, and it will also provide new research ideas for improving gut microbiota therapy for IBD.

## 2. Materials and Methods

### 2.1. Bacterial Strains and Growth Conditions

The *Xanthomonas campestris* pv. *campestris (Xcc)* strain was cultured in Nutrient Agar (NA) medium (5 g/L, 3 g/L beef extract, 10 g/L sucrose, and 1 g/L yeast extract, pH 7.0) at 30 °C. *Staphylococcus aureus* (*S*. *aureus*) (ATCC25923) was cultured in LB medium (tryptone 10 g/L, yeast powder 5 g/L, sodium chloride 10 g/L) at 37 °C for 24 h; *Aeromonas hydrophila* (*A*. *hydrophila*) (ATCC7966) was cultured in Nutrient Broth (NB) medium (peptone 10 g/L, beef meal 3 g/L, sodium chloride 5 g/L, pH 7.2) (Hope Bio-Technology, Qingdao, China) at 30 °C for 24 h.

### 2.2. Preparation of DSF

DSF was prepared as previously described by He et al. [22]. Briefly, the *Xcc* strain was cultured in NA medium at 30 °C for 48 h and centrifuged at 3800 rpm and 4 °C for 30 min to obtain the bacterial supernatant (J6-HC Centrifuge, BECKMAN COULTER™, Brea, CA, USA), and the supernatant pH was adjusted to 4.0. Extraction was carried out twice by the addition of an equal volume of ethyl acetate. The crude DSF was dissolved in methanol and analyzed by high-performance liquid chromatography (HPLC) (Agilent, Palo Alto, CA, USA) with a C18 reversed-phase column (5 µm, 4.6 × 150 mm; Zorbax XDB, Palo Alto, CA, USA).

### 2.3. Zebrafish Maintenance and Embryo Handling

Zebrafish (*Danio rerio*) were raised in a circulating water system with temperature maintained between 25 °C and 28 °C, alternating between 14 h of light and 10 h of darkness, and fed three times with brine shrimp. Zebrafish embryos were produced through natural pairing, collected in egg water (5 mM NaCl, 0.17 mM KCl, 0.33 mM CaCl_2_, and 0.33 mM MgSO_4_, pH = 7.2), and cultured in a constant temperature incubator at 28 °C [23]. Embryos were staged according to the description by Kimmel et al. [24]. This study used the following zebrafish strains: AB wild-type (wt) strain and neutrophil transgenic strain *Tg*(*mpx: eGFP*) [25]. All experimental operations related to zebrafish complied with the ethical guidelines of the Northwest University Laboratory Animal Management and Ethics Committee, and the ethical code was NWU-AWC-20211101Z.

### 2.4. Construction of DSS-Induced Inflammatory Bowel Disease Zebrafish Model

A DSS-induced intestine injury model was constructed as previously described [26,27] with modification. Fresh 0.25% (*w*/*v*) DSS (36,000–50,000 MW, MeilunBio, Dalian, China) was prepared to induce a medium level of enterocolitis in zebrafish larvae at 3 days post-fertilization (dpf). The concentrations of DSF treatment for zebrafish embryos were selected according to the previous description [21]. Since the DSF concentration above 20 μM resulted in developmental toxicity in zebrafish embryos, we chose 5 μM, 10 μM and 20 μM concentrations of DSF for the experiments. Groups were set up as follows: control group, DSS model group (immersed in 0.25% DSS at 3 dpf), and DSF treatment group (immersed in different concentrations of DSF with 0.25% DSS at 3 dpf or immersed in 0.25% DSS at 3 dpf for 2 days followed by the addition of different concentrations of DSF for 2 more days of treatment). All zebrafish embryos were incubated at 28.5 °C, and the culture medium was changed daily. The analyses were carried out at 7 dpf unless otherwise noted. Each group was repeated with 3 wells, 15 larvae per well, and 6 larvae were randomly selected for statistical analysis unless mentioned.

### 2.5. Histological Analysis

#### 2.5.1. Alcian Blue Staining

DSS-induced abnormalities in zebrafish intestinal goblet cells were detected by Alcian blue staining. Zebrafish larvae were collected at 7 dpf, fixed overnight in 4% paraformaldehyde (PFA) at 4 °C, and washed with PBS, and 0.1% Alcian blue staining solution (Sigma, St. Louis, MO, USA) was added for staining overnight at room temperature. After the staining solution was removed, staining samples were washed in acidic alcohol (70% ethanol/1% hydrochloric acid) repeatedly and stored in 50% glycerol for photography using a Nikon SMZ25 microscope system (Nikon, Tokyo, Japan). At least six larvae were examined under each treatment condition.

#### 2.5.2. Hematoxylin–Eosin Staining

In order to detect histopathological changes in the zebrafish intestine, embryos developed to 7 dpf were fixed overnight in 4% PFA at 4 °C. The fixed embryos were dehydrated and embedded in paraffin according to standard protocol. Wax sections of 5 μM were obtained using a LEICA RM2016 slicing machine (LEICA, Wetzlar, Germany). After staining with hematoxylin–eosin (H&E) staining solution, photos were taken using an upright microscope with a digital camera (Nikon, Japan).

### 2.6. Inflammatory Response Analysis

#### 2.6.1. Analysis of Macrophage Accumulation by Neutral Red Staining

Zebrafish larvae at 7 dpf were placed into 6-well plates and mixed with Neutral Red (NR) staining solution (Solarbio, Beijing, China). The larvae were stained for 7 h protected from light at 28.5 °C and then washed with PBS. Finally, the migration and accumulation of macrophages in zebrafish larvae were observed under a Nikon SMZ25 microscope and photographed [21].

#### 2.6.2. Analysis of Neutrophilic Infiltration by Live Imaging

Drug-treated transgenic *Tg*(*mpx:eGFP*) larvae at 7 dpf were anesthetized with 100 μg/mL tricaine for 30 s and mounted in 5% (*w*/*v*) methylcellulose for live imaging with a fluorescent microscope [25]. The level of inflammation was represented by the number of neutrophils clogged in the intestine, which was enumerated by manual counting, and imaged using a Nikon SMZ25 stereoscope (Nikon, Tokyo, Japan).

### 2.7. Microbial DNA Extraction and 16S rRNA Gene Sequencing

Zebrafish larvae from each group (control, DSS, DSS+DSF) were collected, frozen in liquid nitrogen, and stored at −80 °C. Microbial genomic DNA was extracted by magnetic bead method using a Soil and Fecal Genomic DNA Extraction Kit (TianGen, Beijing, China). The V3–V4 region of the bacterial 16S rRNA gene was amplified by PCR using the forward primer F (5′-CCTAYGGGRBGCASCAG-3′) and the reverse primer R (5′-GGACTACNNGGGGTATCTAAT-3′). The library construction was performed using the NEB Next^®^ Ultra™ II FS DNA PCR-free Library Prep Kit (New England Biolabs, Ipswich, MA, USA), and sequencing was performed using the Illumina NovaSeq 6000 platform with PE250 (Novogene, Beijing, China).

The ASVs (amplicon sequence variants) were obtained by concatenating and filtering raw data using FLASH (version 1.2.11) [28] (Appendix A). The sequences were annotated with 100% similarity using DADA2 clustering and QIIME2 (version 2017.6.0) annotation [29]. To compare microbial communities among different samples and groups, stacked bar charts were drawn using the R packages ggplot2 (version 3.4.4) and VennDiagram (version 1.7.3) to display species distribution levels at the phylum and genus levels, as well as the number of microbiotas among different groups. The α-diversity of microbial communities among different groups was evaluated using the vegan package (version 2.6-4) to calculate the Chao1 index for richness and the Shannon index for diversity. The above indices were plotted into a box plot using ggplot2. The significance of the differences was verified through the Kruskal–Wallis rank sum test and Dunn’s test as post hoc tests. The β-diversity of the samples was measured using the vegan package to estimate the difference in community structure between samples. Dimensionality reduction of multidimensional microbial data was carried out using unconstrained sorting methods such as principal coordinate analysis (PCoA) [30] and non-metric multidimensional scaling (NMDS) [31]. By analyzing the distribution of samples on the continuous sorting axis, the main trends in data changes between each group were displayed. LEfSe (LDA effect size) [32] analysis was completed through microeco (version 1.2.0), which combined non-parametric Kruskal–Wallis and Wilcoxon rank sum tests with linear discriminant analysis (LDA) effect size to explore the significance of species composition and community structure differences between samples. LDA value distribution histograms were plotted using ggplot2, GraPhlAn (version 0.9) [33], and pheatmap (version 1.0.12), respectively.

### 2.8. Real-Time Quantitative PCR (qRT-PCR)

Zebrafish larvae were collected, and TRIzolTM reagent (Ambion, Austin, TX, USA) was used for total RNA extraction. Then, 1 μg of total RNA was subjected to cDNA synthesis by a SuperScriptIII^TM^ system (Invitrogen, Carlsbad, CA, USA). qRT-PCR was performed using SYBR Green PCR Master Mix (Kapa Biosystems, Boston, MA, USA) and a ViiA 7 Real-Time PCR System (ABI, Foster City, CA, USA) as previously described [34]. The PCR condition was as follows: 95 °C for 3 min, followed by 40 cycles of 95 °C for 3 s, 60 °C for 20 s. The primer sequences are shown in Appendix A.

### 2.9. Infecting Zebrafish Embryos with A. hydrophila or S. aureus

The infection of zebrafish embryos by *A. hydrophila* or *S. aureus* was performed according to the method of Murugan [35]. In brief, the bacterial stock solution was centrifuged at 3500 rpm for 10 min at 4 °C, and the bacterial suspension was prepared to reach a final experimental concentration of 1 × 10^8^ CFU/mL of *A. hydrophila* or 1 × 10^5^ CFU/mL of *S. aureus*. Zebrafish embryos at 4 dpf (AB or *Tg*(*mpx:eGFP*) line) were selected and placed in 6-well plates containing E3 medium. The experiment was divided into three groups: control group, *A. hydrophila* or *S. aureus* infection group (immersed in *A. hydrophila* or *S. aureus*), and DSF-treated group (immersed in 20 μM DSF with *A. hydrophila* or *S. aureus*). After 6 h of bacterial exposure and treatment, the embryos were taken for further analysis.

### 2.10. Quantitative Analysis of A. hydrophila or S. aureus in the Gut of Zebrafish

A bacterial stock solution of *A. hydrophila* or *S. aureus* was incubated with 5 μM of the molecular probe 5(6)-CFDA, SE, for 20 min at 37 °C for bacterial fluorescence labeling. The residual fluorescent probe on the bacterial surface was washed away using PBS, and bacterial suspensions were prepared to achieve final experimental concentrations of 1 × 10^8^ CFU/mL for *A. hydrophila* or 1 × 10^5^ CFU/mL for *S. aureus*. The 4 dpf zebrafish larvae were placed in 6-well plates, and the experiments were divided into two groups: the infected group (immersed in *A. hydrophila* or *S. aureus*) and the DSF-treated group (immersed in 20 μM DSF with *A. hydrophila* or *S. aureus*). Imaging was performed using a Nikon SMZ25 stereomicroscope after 6 h of treatment at 28 °C, and intestinal fluorescence intensity (IFI) was analyzed using Image-J software (version 1.8.0) [36].

### 2.11. Statistical Analysis

Each experiment was repeated at least three times, and all data were expressed using means ± SD. Comparisons between different groups were performed using Student’s *t*-test. *p* < 0.05 was considered a statistically significant difference.

## 3. Results

### 3.1. DSF Alleviates DSS-Induced Intestinal Damage in Zebrafish

The chemical structure of DSF is shown in Figure 1A. To investigate the influence of DSF on the zebrafish intestine, a zebrafish colitis model was established using DSS (Figure 1B). At 7 dpf, compared with the control group, the addition of DSS resulted in a significant shortening of intestinal length (Figure 1C,C’), indicating that DSS successfully induced intestinal abnormalities in zebrafish larvae. To further confirm the intestinal damage caused by DSS, the changes in goblet cells in zebrafish intestines were observed using Alcian blue staining. As shown in Figure 1D,D’, the number of goblet cells in the DSS-induced group significantly increased by 23% compared to the control group, indicating intestinal mucosal damage. Moreover, histopathological analysis showed that adding DSS to zebrafish embryos also caused significant dilatation of the intestinal lumen and loosening of the tissue structure (Figure 1E,E’). This indicated that the DSS group exhibited significant colitis characteristics in the intestine. After treatment with different concentrations (5 μM, 10 μM, 20 μM) of DSF, the above indicators showed varying degrees of protection in zebrafish with DSS-induced colitis, including an increase in intestinal length (Figure 1C,C’), a decrease in the number of goblet cells to the original level (Figure 1D,D’), and a more complete intestinal tissue structure (Figure 1E). These results suggested that DSF has a preventive effect on DSS-induced colitis in zebrafish.

### 3.2. DSF Inhibits DSS-Induced Inflammation in Zebrafish

In the early stage of development, zebrafish predominantly depend on their innate immune system. During this stage, the principal immune inflammatory cells in zebrafish larvae are neutrophils and macrophages. The localization of neutrophils in the intestine was assessed by live imaging of *Tg*(*mpx:EGFP*) zebrafish embryos. It could be observed that compared with the control group, the DSS group had a large accumulation of neutrophils in the intestine, indicating the presence of severe inflammatory cell infiltration in the intestine. On the contrary, different doses of DSF treatment significantly improved neutrophil infiltration in the zebrafish intestine (Figure 2A). In addition, the migration and recruitment behavior of macrophages stained with neutral red in the zebrafish gut of the DSS group exhibited a similar trend. Different doses of DSF could effectively inhibit the abnormal aggregation of macrophages at the inflammatory sites (Figure 2B).

DSS-induced colitis is also accompanied by the activation of inflammatory pathways and upregulation of inflammatory cytokine levels. As shown in Figure 3A, in the DSS group, the expression levels of pro-inflammatory cytokines (*tnfα*, *il6*, *il1b*) and the inflammatory pathway factors (*cox2*, *nfkb1*, *tab1*, *fadd*) were abnormally upregulated. In the meantime, the anti-inflammatory cytokine *il10* was aberrantly downregulated, while DSF treatment could significantly regulate the expression levels of those factors to varying degrees (Figure 3A). Furthermore, as important transmembrane and intracellular tight junction factors, the expression levels of *claudin1* and *occludin1* were abnormally reduced in the DSS group, but significantly upregulated after DSF treatment, indicating that DSF can induce intestinal barrier reinforcement and may contribute to the recovery of chronic colitis (Figure 3B).

### 3.3. DSF Modulates the Intestinal Flora Structure in DSS-Induced Zebrafish Colitis

To investigate the regulatory effect of DSF on the gut microbiota, 16S rRNA sequencing was performed. The Venn diagram was used to describe the changes in ASVs of the three taxa, and the results showed that there was a total of 99 ASVs coexisting in all three groups; 150 ASVs coexisted with the control and DSS group, while 125 ASVs coexisted with the DSS group and DSS + DSF group. These data indicated that the ASV diversity varies among each group (Figure 4A). The α-diversity measured by the abundance index (Chao1) and diversity index (Shannon) exhibited an increase in the DSS group compared with the control group, while it recovered in the DSF treatment group, indicating that DSF treatment may affect the diversity and abundance of gut microbiota (Figure 4B). To evaluate the differences among microbial communities, β-diversity analysis by principal coordinate analysis (PCoA) and non-metric multidimensional scaling (NMDS) was carried out and showed significant clustering separation between the DSS group and other groups. On the other hand, the Unifrac distance between the DSF group and the control group was closer, indicating that DSS induced changes in the original gut microbiota composition, while DSF treatment exhibited a trend of restoring the altered gut microbiota composition to normal levels (Figure 4C). UPGMA clustering tree analysis showed that the bacterial composition in each group was similar at the phylum level, but there was a significant difference in bacterial abundance between the DSS group and the control group (Figure 4D).

Further analysis of the composition of gut microbiota in different groups showed that at the phylum level, DSS induction significantly increased the relative abundance of *Bacteroidota*, *Actinobacteriota*, and *Firmicutes*, while reducing the relative abundance of *Proteobacteria* in the intestine of zebrafish (Figure 5A,B). At the genus level, among the top 20 most abundant genera in the DSS group, the relative abundance of 15 genera increased, while the relative abundance of 5 genera decreased (Figure 5C,D). Among bacterial genera with significant changes in abundance in the DSS group, the relative abundance of *Aeromonas*, *Bacteroides*, *Flavobacterium*, *Staphylococcus*, *Vibrio*, and *Enterococcus* increased, while the relative abundance of *Pseudomonas* and *Plesiomonas* decreased. However, DSF treatment could partially suppress these abnormal changes in bacterial genera abundance caused by DSS (Figure 5E).

### 3.4. LEfSe Analysis of Intestinal Microbiota

Linear discriminant analysis effect size (LEfSe) analysis can display differences in the composition and structure of intestinal microbiota among different groups and identify biomarkers and dominant microbiota among different groups. Through LDA score cladogram and histogram analysis, as shown in Figure 6A,B, from the order to the genus level, it was shown that the dominant species in the control group were *g_Pseudomonas*, *f_Chromobacteriaceae*, and *p_Proteobacteria*, while after the induction of DSS, the dominant species shifted to *f_Aeromonas* and *p_Bacteroidota*. The DSF group showed different dominant species, such as *p_Actinobacteriota* (Figure 6A,B). The species composition heatmap displayed the distribution trend of species abundance in different samples, intuitively reflecting the differences in species composition. Compared with the control group, the species distribution trend of the DSS group at the genus level was different, with significant differences in composition, while the distribution pattern of the DSF group was more similar to that of the control group (Figure 6C).

### 3.5. Inhibitory Effect of DSF on the Proliferation and Infection of A. hydrophila and S. aureus in Zebrafish

Through the intestinal microbiota analysis, it was found that the DSS group had an abnormal increase in *A. hydrophila* and *S. aureus*, which were considered common pathogens of IBD, while DSF treatment could effectively inhibit the increase in the abundance levels of the two pathogenic bacteria in the intestine of DSS-induced colitis. In order to further investigate the direct effect of DSF on intestinal pathogens, 20 µM of DSF was added to culture media containing *A. hydrophila* or *S. aureus*. The bacterial growth curve results showed that DSF had a significant inhibitory effect on the proliferation of *A. hydrophila* and *S. aureus* (Figure 7A,B). By infecting zebrafish with bacteria, quantitative analysis of *A. hydrophila* or *S. aureus* showed that DSF could also inhibit the colonization of two pathogenic bacteria in the gut of zebrafish (Figure 7C,C’,D,D’). Intestinal pathological tissue sections showed that compared to the control group, zebrafish infected with *A. hydrophila* or *S. aureus* exhibited varying degrees of intestinal dilation, loose arrangement of intestinal epithelial cells, and intestinal wall gaps. DSF treatment could effectively reduce these pathological symptoms caused by intestinal infection (Figure 7E,E’,F,F’). Furthermore, compared with the control group, infection with *A. hydrophila* or *S. aureus* significantly increased the number of goblet cells in the zebrafish intestine, which was significantly decreased after DSF treatment (Figure 8A). The abnormal increase in neutrophils and macrophages in the zebrafish intestine caused by infection with *A. hydrophila* or *S. aureus* could be significantly inhibited by DSF (Figure 8B,C). These results further indicated that DSF may achieve preventive effects on colitis by inhibiting the proliferation and infection damage of pathogenic bacteria in zebrafish.

## 4. Discussion

The signal interaction between humans and microbes is constantly occurring. There are numerous, diverse, and complex microbial communities in the human gut that are closely associated with various physiological functions of the host, including growth and development, nutrition and metabolism, immune response, and intestinal barrier protection [37]. The imbalance of the intestinal microbiota is considered a crucial factor in promoting the occurrence and progression of IBD. Therefore, restoring intestinal homeostasis by regulating the gut microbiota is expected to become a treatment approach for IBD [38]. The quorum-sensing signals produced by bacteria regulate the collective behavior of microbial populations and communities in the gastrointestinal tract and may affect community composition and host physiology [39]. In our previous study, we found that the QS signal DSF has a therapeutic effect on LPS-induced intestinal inflammatory injury, which may be related to the regulation of the Toll-like receptor signaling pathway [21]. In the present study, using a DSS-induced zebrafish colitis model, we found that DSF could improve the symptoms of colitis, regulate the expression of inflammatory factors, regulate goblet cell abnormalities, enhance intestinal tight junctions, and more importantly, modulate the composition of the host gut microbiota.

The interaction between the intestinal epithelial barrier (IEB) and the gut microbiota and mucosal immune system plays a crucial role in maintaining human health [40]. The mucus layer on the surface of the intestine is the first line of defense against mechanical, chemical, and biological damage. Its main function is to protect epithelial cells from external harmful substances, which is also critical for maintaining intestinal homeostasis [40,41]. Here, in the DSS-induced zebrafish IBD model, the mucus layer in the zebrafish intestine was damaged, which exacerbated the development of colitis. The mucus layer is mainly continuously secreted by goblet cells in the intestine. Under normal conditions, goblet cells accumulate in the midgut of zebrafish [42]. In the IBD model group, an abnormal increase in goblet cell mucus secretion was observed. DSF treatment significantly inhibited the increase in intestinal mucus and repaired the damage to the intestinal mucosal barrier. Tight junctions are a type of intercellular connection that is crucial for establishing epithelial barriers and maintaining epithelial polarity. The dynamic equilibrium between tight junctions and gut microbiota is vital for maintaining intestinal homeostasis [43]. Compared with the normal group, we detected a significant decrease in the expression of tight-junction-related genes (*claudin1*, *occludin1*) in the zebrafish IBD model group, while DSF treatment significantly restored the abnormal upregulated expression of *claudin1* and *occludin1*. These results indicate that DSF has a protective effect on the intestinal epithelial barrier damage induced by DSS in zebrafish, which is important for maintaining the intestinal structure and microbial environment.

Persistent inflammation is a prominent feature of colitis, and inhibiting inflammation has become an important therapeutic approach for alleviating colitis [44]. The destruction of intestinal epithelium can lead to immune dysfunction and trigger inflammatory reactions, which are considered the main immune factors in the occurrence of IBD. According to previous reports, inflammatory cytokines such as TNF-α, IL-1β, and IL-6 are closely associated with the occurrence of intestinal inflammation [45]. Here, in the zebrafish colitis model, DSF treatment significantly inhibited the abnormal increase in the expression of pro-inflammatory cytokines (*tnfa*, *il1b*, and *il6*) and the decrease in the expression of the anti-inflammatory factor *il10*. IL-10 is a key immunosuppressive cytokine that regulates intestinal homeostasis, and its restriction of a pro-inflammatory response plays an important role in reducing the severity of colitis [46]. Increased expression of IL-10 is also beneficial for restoring intestinal barrier function [47]. The imbalance of microbial homeostasis leads to the overgrowth of harmful bacteria, which produce virulence factors that penetrate the intestinal barrier and are recognized by innate immune cells through conservative pattern recognition receptors, thereby stimulating intestinal inflammatory response. This process involves the activation of related inflammatory pathways, such as the NF-κB pathway [48,49]. We found that DSF treatment could inhibit an abnormal increase in the expression of NF-κB-pathway-related genes (*cox2*, *nfkb1*, *tab1*, *fadd*), thereby alleviating intestinal inflammation. Therefore, DSF participates in regulating pro-inflammatory cytokines and the NF-κB inflammatory pathway to inhibit intestinal inflammatory responses and repair the intestinal barrier, thereby effectively alleviating the symptoms of colitis.

The gut microbiota is a key bridge between environmental factors and host health. According to their impact on the human body, gut microbes can be divided into probiotics, neutral bacteria, and harmful bacteria [8]. DSS-induced colitis can cause an imbalance of gut microbiota and overgrowth of harmful bacteria. Harmful bacteria can directly disrupt the intestinal barrier by secreting toxins and invading intestinal mucosal cells, leading to mucosal inflammation. Harmful bacteria can also stimulate the excessive release of inflammatory cytokines from intestinal epithelial cells and immune cells, causing dysregulation of humoral and cellular immune responses and resulting in persistent inflammatory damage to the intestinal mucosa [50]. In addition, harmful bacteria can also inhibit the growth of probiotics by competing for adhesion sites, secreting inhibitors, disrupting the balance of intestinal microecology, and weakening the structural integrity and immune defense function of the intestine [51].

In the present study, analysis of the gut microbiota showed that the dysbiosis of gut microbiota in the DSS-induced zebrafish IBD model was positively regulated after treatment with DSF. DSF significantly downregulated the abundance of *Bacteroides*, which is a symbiotic bacterium in the intestines. Normally, there is a reciprocal symbiotic relationship between *Bacteroides* and hosts. However, in IBD patients, due to increased permeability of intestinal epithelial cells and disruption of epithelial barrier function, infiltration of *Bacteroides* can lead to abnormal mucosal immune response, and in some cases, infection can lead to sepsis [52]. In addition, it was found in the DSS-induced zebrafish model, the relative abundance of *Aeromonas* and *Staphylococcus*, which are closely related to the development of colitis, also increased. An increase in the relative abundance of *Aeromonas* was observed in patients with inflammatory bowel disease, including ulcerative colitis and Crohn’s disease. Its infection symptoms include diarrhea, bleeding, inflammation, and abdominal pain in IBD [53]. *Staphylococcus* is a type of pathogenic Gram-positive bacteria with a high detection rate in various inflammatory wounds. Peptidoglycan, which is present in the cell wall of *Staphylococcus*, is an immunostimulatory component that can promote the occurrence of inflammatory reactions. In IBD patients, *S. aureus* is more prone to co-infection with other bacterial pathogens, thereby affecting the process of treating IBD [54,55]. DSF treatment resulted in a significant decrease in the relative abundance of these pathogenic bacteria associated with the development of colitis. This indicates that the treatment of DSS-induced colitis with DSF may be related to its inhibition of the proliferation of harmful bacteria in the intestine. At present, antibiotic therapy is commonly used to treat IBD caused by bacterial infections. However, its broad-spectrum activity can cause disruption of the gut microbiota and its colonization resistance to pathogenic microorganisms [56,57]. Using DSF as an adjunct to antibiotic therapy may be a solution to these problems.

In summary, the results of this study indicate that DSF can suppress the DSS-induced colitis phenotype in zebrafish, including by inhibiting intestinal mucus layer proliferation, reducing intestinal shortening, and inhibiting intestinal inflammatory cell aggregation. In addition, DSF regulates the expression of inflammatory cytokines and intestinal tight junction genes, as well as the activation of the NF-κB inflammatory pathway, which not only reduces the immune inflammatory response in the intestine but also enhances the repair of the intestinal mucosa. In terms of gut microbiota, DSF can protect the relative abundance and diversity of gut microbiota under colitis and inhibit the colonization and infection of intestinal pathogens such as *A. hydrophila* and *S. aureus* in the host intestine (Figure 9).

Although DSF has shown good preventive effects in zebrafish colitis models, as aquatic animals, zebrafish have a significant structural difference in gut microbiota compared to terrestrial mammals. In this study, some common bacterial pathogens reported in mammalian colitis were not detected in the gut of zebrafish, possibly due to differences in their surrounding living environments. Whether DSF also has inhibitory effects on other common colitis pathogens remains to be further studied in mammalian models.

## 5. Conclusions

This study indicates that DSF, as a quorum-sensing molecule, can significantly reduce the pathological characteristics of colitis and has great potential in regulating cytokines and inflammatory pathways, as well as preventing gut microbiota imbalance. This study provides a reference for the use of DSF as a dietary supplement for alleviating colitis and as an adjuvant antibiotic treatment.

## Figures and Tables

**Figure 1 nutrients-16-01562-f001:**
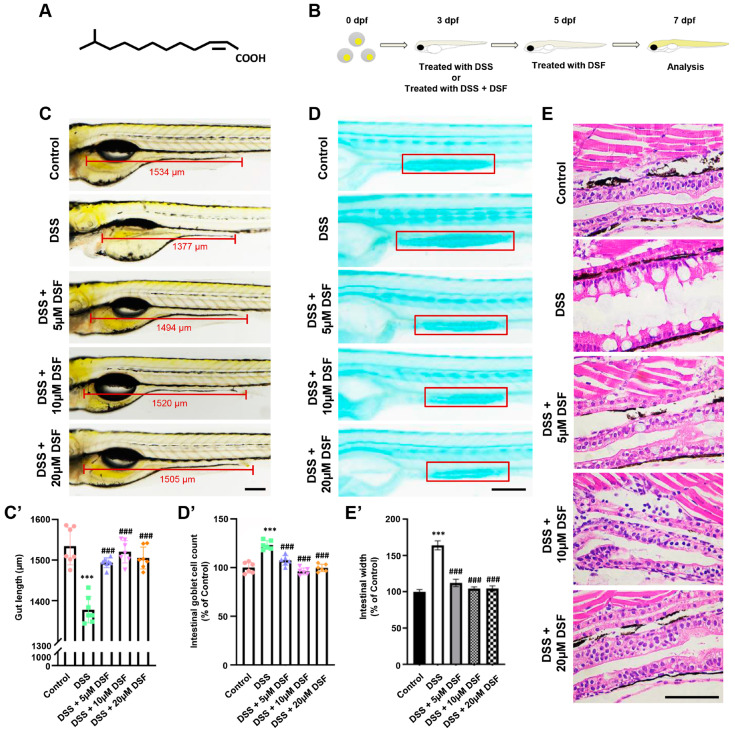
DSF ameliorated DSS-induced intestinal damage in zebrafish larvae. (**A**) The chemical structure of DSF. (**B**) Schematic diagram of the zebrafish experimental design. (**C**,**C’**) Representative images and the data of gut length from different groups. (**D**,**D’**) Detection and counting of goblet cells in zebrafish (red box) using Alcian blue staining method. (**E**,**E’**) Representative images and the data of histopathological examination of zebrafish intestine using H&E staining. Data are shown as mean ± S.D. *** *p* < 0.001 vs. control group; ^###^ *p* < 0.001 vs. DSS group. Scale bar, 100 μm.

**Figure 2 nutrients-16-01562-f002:**
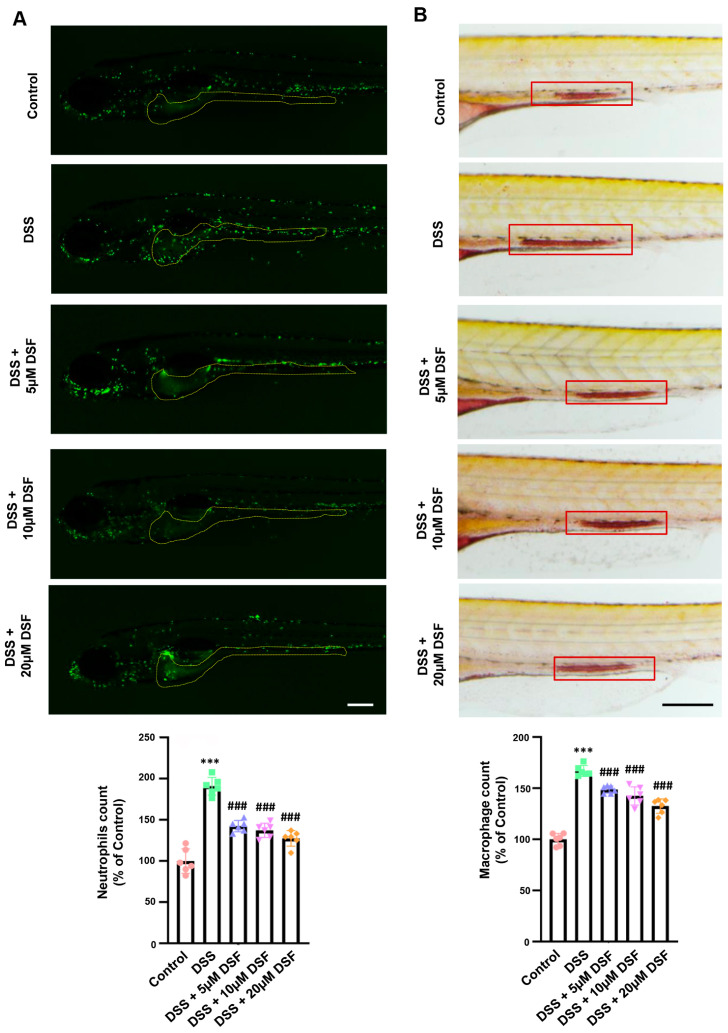
DSF inhibited the aggregation of neutrophils and macrophages in the inflammatory sites of zebrafish induced by DSS. (**A**) Live imaging of *Tg*(*mpx:eGFP*) larva and quantitative analysis of neutrophils in the yellow dotted line areas. (**B**) Neutral red staining showed macrophage accumulation (red boxes) and the quantitative analysis of macrophages in the red box areas. Data are shown as mean ± S.D. *** *p* < 0.001 vs. control group; ^###^ *p* < 0.001 vs. DSS group. Scale bar, 100 μm.

**Figure 3 nutrients-16-01562-f003:**
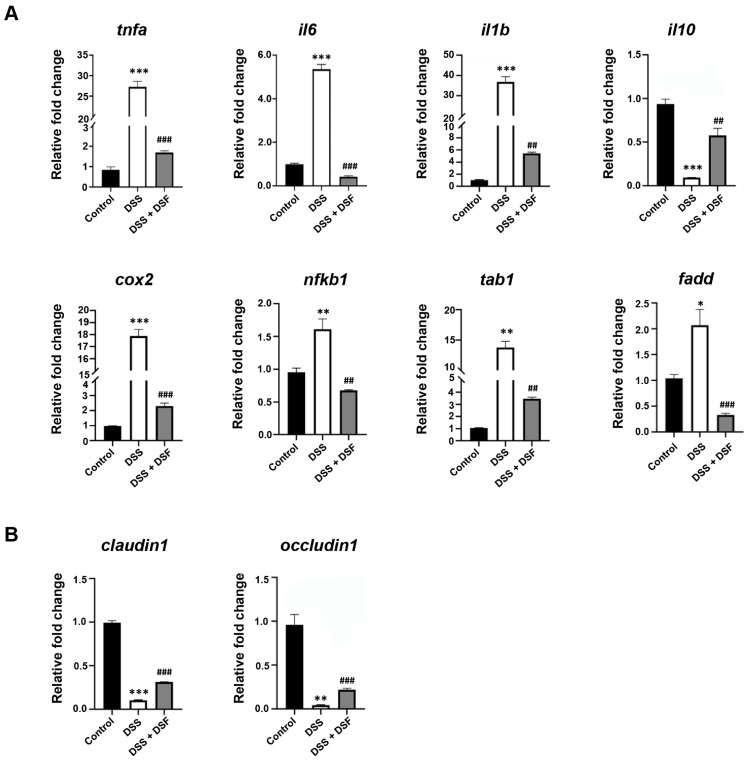
DSF regulates the expression of inflammatory factors and tight junction markers. (**A**) The relative mRNA expression levels of pro-inflammatory cytokines (*tnfα*, *il6*, *il1b*, *il10*) and inflammatory pathway factors (*cox2*, *nfkb1*, *tab1*, *fadd*) were measured with qRT-PCR. (**B**) The relative mRNA expression levels of tight junction markers (*claudin1*, *occludin1*). Data are shown as mean ± S.D. * *p* < 0.05, ** *p* < 0.01, *** *p* < 0.001 vs. control group; ^##^
*p* < 0.01, ^###^ *p* < 0.001 vs. DSS group.

**Figure 4 nutrients-16-01562-f004:**
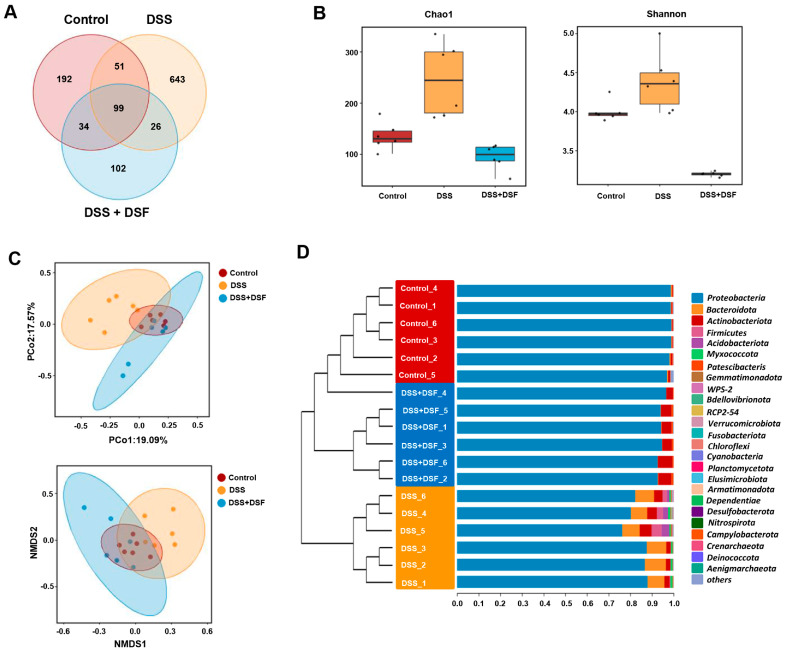
DSF regulates the diversity of gut microbiota in zebrafish colitis induced by DSS. (**A**) Venn diagram indicating the species richness determined by ASVs in each group. (**B**) Evaluation of α-diversity, as determined by abundance index (Chao1) and diversity index (Shannon). (**C**) Evaluation of β-diversity, as determined by PCoA and NMDS analysis. (**D**) The UPGMA clustering tree analysis. *n* = 6 for each group.

**Figure 5 nutrients-16-01562-f005:**
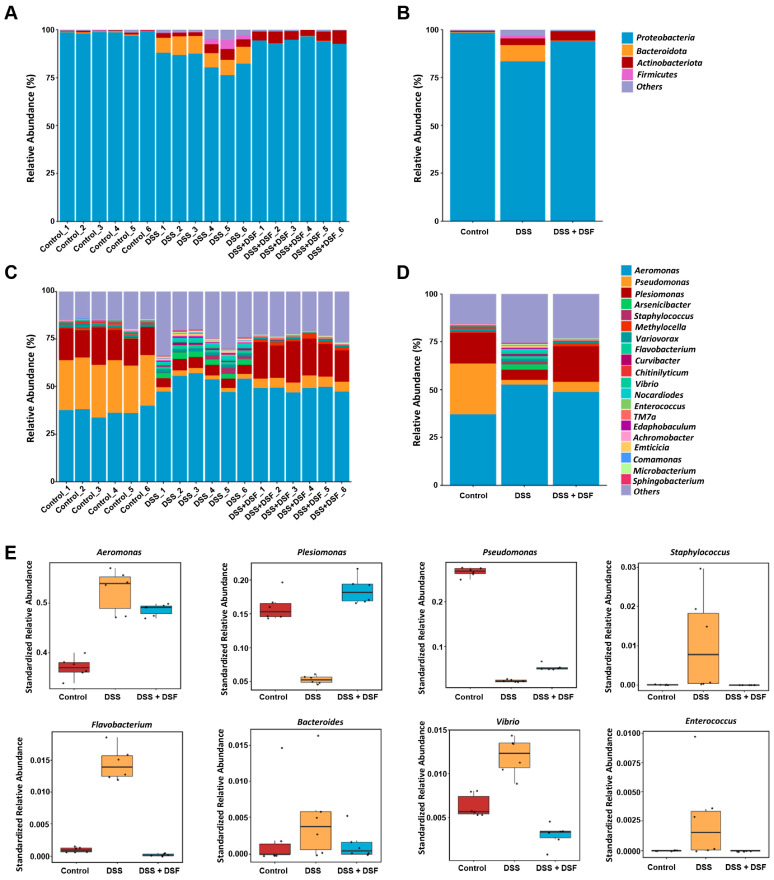
Effects of DSF on species composition. Relative abundance plots displaying the differences in the microbial community structure at the phylum level (**A**,**B**) and genus level (**C**,**D**). (**E**) Genera of bacteria with significant changes related to inflammation. *n* = 6 for each group.

**Figure 6 nutrients-16-01562-f006:**
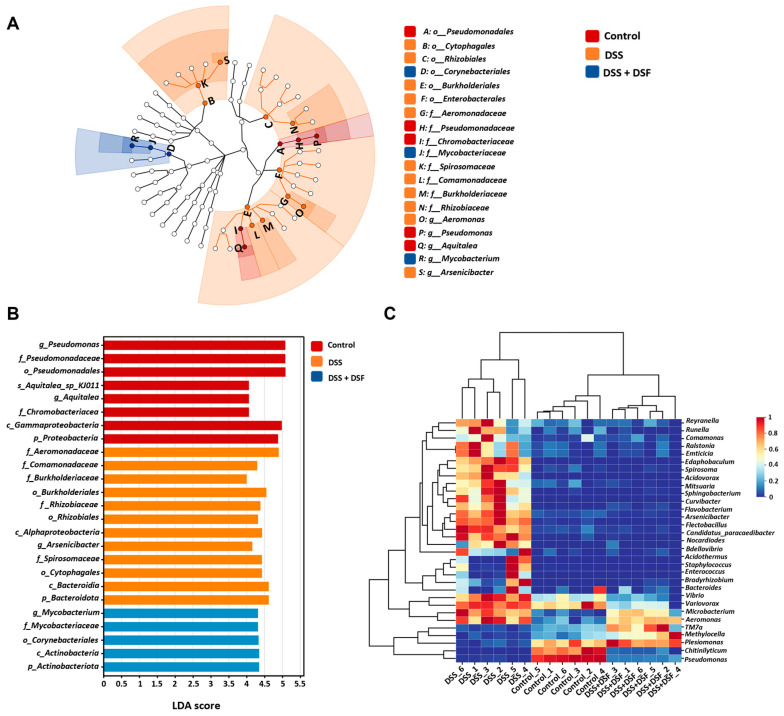
Microbial taxa discrepancies under the effect of DSF. LEfSe analysis of cladogram (**A**) and histogram (**B**). (**C**) Heatmap analysis at the genus level. *n* = 6 for each group.

**Figure 7 nutrients-16-01562-f007:**
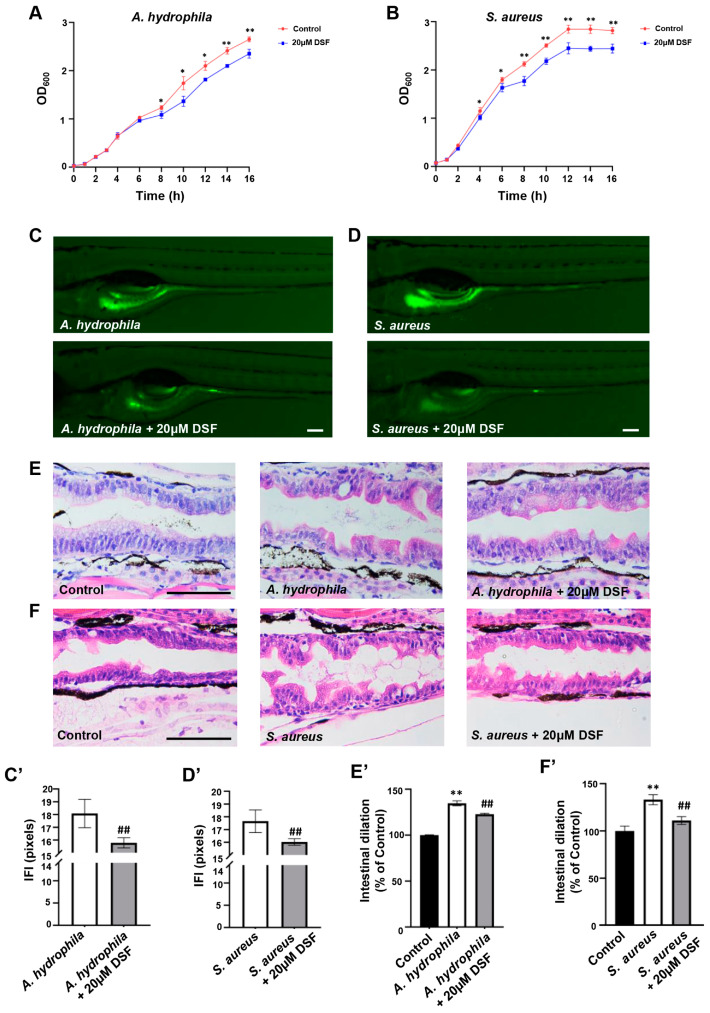
DSF suppressed the infection of *A. hydrophila* or *S. aureus* in zebrafish. Growth curves of *A. hydrophila* (**A**) and *S. aureus* (**B**) under 20 µM DSF treatment. Fluorescence images (**C**,**D**) and quantitative analysis of intestinal fluorescence intensity (**C’**,**D’**) of *A. hydrophila* or *S. aureus* in zebrafish intestine. Histopathological examination of zebrafish intestine infected with *A. hydrophila* (**E**,**E’**) or *S. aureus* (**F**,**F’**). Data are shown as mean ± S.D. * *p* < 0.05, ** *p* < 0.01 vs. control group; ^##^
*p* < 0.01 vs. *A. hydrophila* group or *S. aureus* group. Scale bar, 100 μm.

**Figure 8 nutrients-16-01562-f008:**
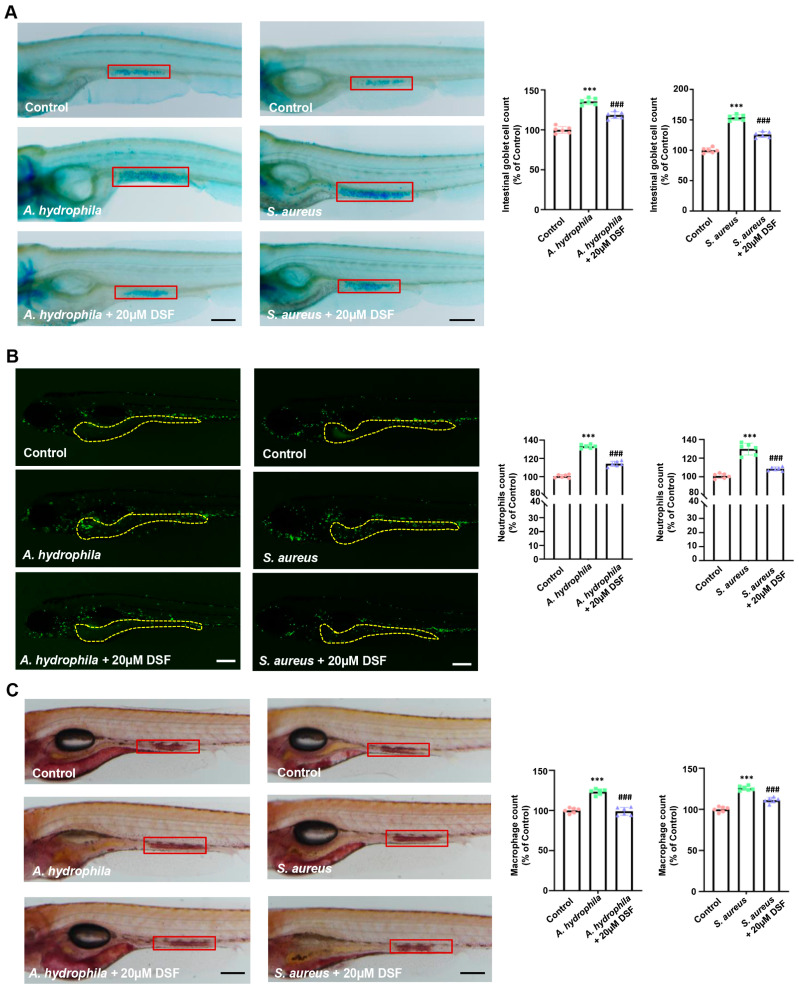
DSF restrained intestinal damage caused by the infection of *A. hydrophila* or *S. aureus* in zebrafish. (**A**) Imaging of Alcian blue staining and counting of goblet cells in the intestine of zebrafish (red box). (**B**) Live imaging of *Tg*(*mpx:eGFP*) zebrafish larva infected with *A. hydrophila* or *S. aureus* and quantitative analysis of neutrophils. (**C**) Imaging of macrophage accumulation (red boxes) and the quantitative analysis of macrophages in the intestine of zebrafish (red box). Data are shown as mean ± S.D. *** *p* < 0.001 vs. control group; ^###^ *p* < 0.001 vs. *A. hydrophila* group or *S. aureus* group. Scale bar, 100 μm.

**Figure 9 nutrients-16-01562-f009:**
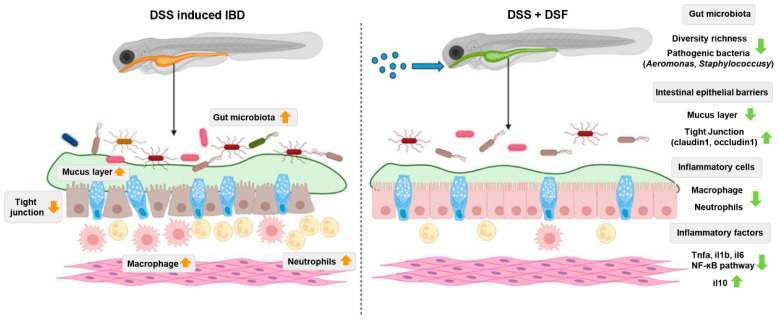
Schematic diagram of the molecular mechanism of DSF improving DSS-induced zebrafish colitis.

## Data Availability

The data presented in this study are available on request from the corresponding author due to privacy reasons.

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
