# Peer review of "Quorum-Sensing Signal DSF Inhibits the Proliferation of Intestinal Pathogenic Bacteria and Alleviates Inflammatory Response to Suppress DSS-Induced Colitis in Zebrafish"

_nutrients, 2024, doi:10.3390/nu16111562_

Round 1
Reviewer 1 Report
Comments and Suggestions for Authors
In their manuscript, Yi et al. describe an IBD model in zebrafish and the treatment using DSF. The work was done very well and the manuscript describes it well. The results are interesting and should be published. A few minor and one major points should be adressed.
1. What is the Xcc strain.
2. in Fig. 7. in the DSF panels need to be specified that S. hydrophila and S. aureus was also added.
A major point is that the DSF was given at the same time as DSS or the bacteria. Thus it it is a preventive strategy and not a therapeutic one. This is acceptable in a first publication but a therapeutic effect after established IBD needs to be tested later. However, this should be considered in the text. Thus, words like restored or rescued etc. should be avoided since the activity of DSS and bacteria is only inhibited.
Author Response
Response to Reviewer 1 Comments
Point 1: What is the Xcc strain.
Response 1: Thank you for pointing out the issue. The Xcc strain is the plant pathogen Xanthomonas campestris pv. campestris (Xcc), which synthesizes and secretes the quorum-sensing signal (Diffusible signal factor, DSF), we have indicated the full name in the manuscript (Line 99).
Point 2: in Fig. 7. in the DSF panels need to be specified that S. hydrophila and S. aureus was also added.
Response 2: Thank you for pointing out the oversight in Fig.7. We have changed the DSF panels in Fig. 7 to A. hydrophila+20µM DSF, S. aureus+20µM DSF.
Point 3: A major point is that the DSF was given at the same time as DSS or the bacteria. Thus it it is a preventive strategy and not a therapeutic one. This is acceptable in a first publication but a therapeutic effect after established IBD needs to be tested later. However, this should be considered in the text. Thus, words like restored or rescued etc. should be avoided since the activity of DSS and bacteria is only inhibited.
Response 3: Thank you very much for your valuable suggestion. It is right to change the therapeutic strategy mentioned in the manuscript to a preventive strategy, and we have revised all the expressions with therapeutic connotations in the manuscript.
Reviewer 2 Report
Comments and Suggestions for Authors
Yi et al assess the influence of the quorum sensing (QS) molecule cis-11-methyldecenoic acid (DSF) on DSS-induced experimental colitis in zebrafish. The impact of QS molecules on the intestinal environment in health and during disease is intriguing, with the authors showing that DSF appears to attenuate colon shortening and neutrophil infiltration. The authors also demonstrate that DSF may influence the composition of the microbiota, notably the pathogens A. hydrophilia and S. aureus.
This work represents a good start to a promising manuscript but is critically lacking in description of the experimental procedures, histological analyses, and experiments that support the claims of improved intestinal barrier permeability.
Comments are as follows:
Line 91: 16S sequencing only gives composition of the microbiota, not function / mechanism.
Figure 1D and D’: How was the number of goblet cells quantitated? Were the number of cells counted? The number of goblet cells cannot be reliably inferred from the amount of mucin, if this is the method the authors are using. Zoom into region of interest on Figure 1D
A’, B’, C’, D’, E’, and F’ on figures is not necessary.
Lines 241 – 249 (Figure1 E), and Lines 362 – 365 (Figures 7C and D): Histological analyses need quantitation to support claims. Scoring to assess damage, inflammation, etc., and quantitation of dilation (width of lumen).
Label bar lengths in Figure 1C
Figure 2A is difficult to see, zoom into region of interest
How were macrophages quantitated for Figure 2B’?
Figures 7A and B: what is the level of inhibition at 5 uM? As 5 uM shows a similar protective effect against DSS-induced colitis as 20 uM, if inhibition of pathogenic growth is a primary mechanism of protection, 5 uM should also similarly limit growth. A DSF dose-response experiment is recommended.
Figures 7E and F are difficult to see, draw a line around intestine as done in Figure 2A and zoom into that region.
A. hydrophilia and S. aureus levels need quantitation in infection experiments.
Line 392: Improvement of intestinal barrier function cannot be inferred by expression of junctional components, intestinal permeability assays must be performed.
How was histology collected / fixed etc.?
Intestinal epithelial apoptosis and proliferation analyses would be beneficial.
Why weren't goblet cells and macrophages assessed in infection experiments as well?
Comments on the Quality of English Language
The English is confusing in some parts, the manuscript needs editing for clarity.
Author Response
Please also check the upload word file for the response
Response to Reviewer 2 Comments
Point 1: 16S sequencing only gives composition of the microbiota, not function / mechanism.
Response 1: Thank you for pointing out the inappropriate description in the manuscript, we have changed the original text to “Meanwhile, using 16s rRNA sequencing, we investigated the regulation of DSF on structure and composition of colonic microbiota in colitis from the perspective of the gut microbiota.”(Line 91).
Point 2: Figure 1D and D’: How was the number of goblet cells quantitated? Were the number of cells counted? The number of goblet cells cannot be reliably inferred from the amount of mucin, if this is the method the authors are using. Zoom into region of interest on Figure 1D
Response 2: Thank you for your question. Previous research reports have stated that ”Whole mount alcian blue staining is a useful technique for visualizing the acidic mucin produced by zebrafish goblet cells. Furthermore, stereomicroscopy of alcian blue-stained whole mount specimens is sufficient to monitor mid-intestinal goblet cell density and DSS-induced intestinal mucus accumulation”(Oehlers, Stefan H et al. “Chemically induced intestinal damage models in zebrafish larvae.”), in other research, the staining results of Alcian blue were also directly used to represent the number of goblet cells in the midgut(Chen, Yi-Chung et al. “Zebrafish Agr2 is required for terminal differentiation of intestinal goblet cells.”). Therefore, we used Alcian blue staining to display goblet cells in the intestine (stained particles representing goblet cells), and enlarged the corresponding part of the figure. We counted the number of stained particles in the intestinal area. When displaying the quantitative results of goblet cells, we used the percentage of each group relative to the control group to represent it.
Point 3: A’, B’, C’, D’, E’, and F’ on figures is not necessary.
Response 3: Due to the addition of supplementary experiments and reassignment of figures layout in the revised manuscript, adding quoted letters is beneficial for readers to understand the figures, thus, we have to use A ', B', C’ D’ E’ labels on the figures.
Point 4: Lines 241 – 249 (Figure1 E), and Lines 362 – 365 (Figures 7C and D): Histological analyses need quantitation to support claims. Scoring to assess damage, inflammation, etc., and quantitation of dilation (width of lumen).
Response 4: Thank you for pointing out the shortcomings of pathological analysis and providing suggestions. We have checked the previous studies, but there seems to be no quantitative method for histological analysis in zebrafish juveniles(Flores, Erika et al. “Colonization of larval zebrafish (Danio rerio) with adherent-invasive Escherichia coli prevents recovery of the intestinal mucosa from drug-induced enterocolitis.”; Huang, Xuedi et al. “A Rapid Screening Method of Candidate Probiotics for Inflammatory Bowel Diseases and the Anti-inflammatory Effect of the Selected Strain Bacillus smithii XY1.”). We have followed your suggestion to measure and quantitatively analyze the width of the intestinal lumens in each group, revised manuscript added new pane Figure 1E’ in Figure 1 and Figure 7E’ F’ in Figure 7 to show the quantitative analysis of width of the intestinal lumens. We hope these results can support our statement.
Point 5: Label bar lengths in Figure 1C
Response 5: Thank you for your suggestions. We labeled the length of the bar in Figure 1C.
Point 6: Figure 2A is difficult to see, zoom into region of interest
Response 6: Thank you for pointing out the shortcomings in the photos. We have enlarged the intestinal area marked by the yellow dashed line in Figure 2A to make the image of neutrophils clearer.
Point 7: How were macrophages quantitated for Figure 2B’?
Response 7: Thank you for your question. Macrophages can be labeled after phagocytosis of neutral red dye. Due to the presence of significant interference from zebrafish melanin in the previous neutral red staining images, we have replaced the images to ensure clearer staining results. We counted and quantified the red macrophages contained in the red box in Figure 2B using the Iamge J software.
Point 8: Figures 7A and B: what is the level of inhibition at 5 uM? As 5 uM shows a similar protective effect against DSS-induced colitis as 20 uM, if inhibition of pathogenic growth is a primary mechanism of protection, 5 uM should also similarly limit growth. A DSF dose-response experiment is recommended.
Response 8: Thank you for your suggestions. We have added 5 µM and 10 µM DSF for dose response experiments on Aeromonas hydrophila and Staphylococcus aureus, as shown in the the below, it can be seen that 5 µM and 10 µM DSF can also inhibit the proliferation of A. hydrophila and S. aureus, while 20 µM DSF has the most significant inhibitory effect.
Growth curves of A. hydrophila and S. aureu under gradient dose of DSF treatment. (A)At 8 hours of cultivation, the proliferation of A. hydrophila is significantly inhibited by treatment with 20 µM DSF; at 14 hours of cultivation, 10 µM DSF shows a significant inhibitory effect on the proliferation of A. hydrophila; and at 16 hours of cultivation, 5 µM DSF significantly inhibits the proliferation of A. hydrophila. (B) At 4 hours of cultivation, the proliferation of S. aureus is significantly inhibited by treatment with 20 µM DSF; at 10 hours of cultivation, 10 µM DSF shows a significant inhibitory effect on the proliferation of S. aureus; and at 14 hours of cultivation, 5 µM DSF significantly inhibits the proliferation of S. aureus.
Point 9: Figures 7E and F are difficult to see, draw a line around intestine as done in Figure 2A and zoom into that region.
Response 9: Thank you for your suggestions. In the revised manuscript, we added a new Figure 8. In Figure 8A, we have marked the intestinal area of zebrafish with lines and enlarged the intestinal area for a clear display of neutrophil images.
Point 10: A. hydrophilia and S. aureus levels need quantitation in infection experiments.
Response 10: To solve this problem, we used fluorescent probes to pre-label A. hydrophila and S. aureus for zebrafish infection experiments. After DSF treatment, we analyzed the level changes of the two pathogens in the zebrafish intestine based on the fluorescence level in the zebrafish intestine. As shown in the revised manuscript, new panels of Figure 7C, C’ and 7D,D’ were added, 20 µM DSF could significantly inhibit the colonization of A. hydrophila and S. aureus in the intestine of zebrafish.
Point 11: Line 392: Improvement of intestinal barrier function cannot be inferred by expression of junctional components, intestinal permeability assays must be performed.
Response 11: Thank you for pointing out th inappropriate narration in the manuscript. In this study, we attempted to detect intestinal goblet cells using Alcian blue staining, detect intestinal pathological damage through intestinal histological analyses, and detect the expression of intestinal tight junction genes to demonstrate the protective effect of DSF on the intestinal barrier in colitis. Unfortunately, we have not yet found a corresponding detection method for intestinal permeability testing in zebrafish. Therefore, we changed the description of enhancing intestinal barrier function to “regulate goblet cell abnormalities and enhance intestinal tight junction” (Line 425).
Point 12: How was histology collected / fixed etc.?
Response 12: In manuscript lines 156-160, we mentioned that the zebrafish was collected after infection and DSF treatment, which at 4dpf. After cleaning the fish body with PBS three times, 4% PFA was added and fixed overnight at 4 ℃. The next day, routine procedures such as gradient alcohol dehydration, xylene transparency, paraffin embedding, sectioning, and HE staining were performed.
Point 13: Intestinal epithelial apoptosis and proliferation analyses would be beneficial.
Response 13: Thank you for your suggestions. Based on the results of HE staining, it was hard to analyze the apoptosis and proliferation of intestinal epithelial cells among the three groups. Therefore, we conducted an additional acridine orange staining to detect the level of apoptosis in the intestine. As shown in the following figure, the results of acridine orange staining did not show significant differences in the apoptosis of intestinal epithelial cells among the groups.
Detection of apoptosis in intestinal epithelial cells. (A, B) Using AO fluorescent staining to detect the apoptosis of zebrafish intestinal cells infected with A. hydrophila (A) or S. aureus (B). (C, D) The fluorescence intensity was quantified for individual zebrafish gut using ImageJ analysis. Scale bar, 100 μm.
Point 14: Why weren't goblet cells and macrophages assessed in infection experiments as well?
Response 14: Thank you for your suggestions. We have supplemented the analysis of goblet cell and macrophages in bacterial infection experiments. As shown in the revised manuscript Figure 8 , the results of Alcian blue staining (Figure 8A) and neutral red staining (Figure 8C) showed that the abnormal increase of goblet cells and macrophages in zebrafish intestines caused by infection with A. hydrophila or S. aureus could be significantly inhibited by DSF

Reviewer 3 Report
Comments and Suggestions for Authors
This manuscript shows very interesting results in relation to host microbe interaction and disease processes. The research is well designed and presented, my major comment is on the use of English language and the clarity of the text. Sometimes is looks like words are missing e.g. in the introduction: ameliorating the shortening of the intestinal, sometimes text is ambiguous: restoring the intestinal pathological damage, does this mean that damage is decreasing or increasing? All through the document comparable examples can be found and this needs clear improvement
Comments on the Quality of English Language
-
Author Response
Response to Reviewer 3 Comments
Point 1: my major comment is on the use of English language and the clarity of the text. Sometimes is looks like words are missing e.g. in the introduction: ameliorating the shortening of the intestinal, sometimes text is ambiguous: restoring the intestinal pathological damage, does this mean that damage is decreasing or increasing? All through the document comparable examples can be found and this needs clear improvement
Response 1: Thank you for pointing out the language usage and text clarity issues in the manuscript and providing us with examples. We are aware that there were many inappropriate expression in our initial draft, and we have reviewed and revised the entire manuscript to identify any unclear expressions and errors and also invited professionals to review the manuscript. Thank you again for your help and suggestions.
Round 2
Reviewer 2 Report
Comments and Suggestions for Authors
Thank you for the detailed and thoughtful addressing of the comments.
Reviewer 3 Report
Comments and Suggestions for Authors
The changes in the manuscript have led to a clearly improved version, please check some small language (especially singular and plural issues) .
Comments on the Quality of English Language
-